# Antifungal and Larvicidal Activities of Phlorotannins from Brown Seaweeds

**DOI:** 10.3390/md19040223

**Published:** 2021-04-16

**Authors:** Bertoka Fajar Surya Perwira Negara, Jae-Hak Sohn, Jin-Soo Kim, Jae-Suk Choi

**Affiliations:** 1Seafood Research Center, Industry-Academic Cooperation Foundation, Silla University, 606, Advanced Seafood Processing Complex, Wonyang-ro, Amnam-dong, Seo-gu, Busan 49277, Korea; ftrnd12@silla.ac.kr (B.F.S.P.N.); jhsohn@silla.ac.kr (J.-H.S.); 2Department of Food Biotechnology, College of Medical and Life Sciences, Silla University, 140, Baegyang-daero 700beon-gil, Sasang-gu, Busan 46958, Korea; 3Department of Seafood and Aquaculture Science, Gyeongsang National University, 38 Cheondaegukchi-gil, Tongyeong-si, Gyeongsangnam-do 53064, Korea

**Keywords:** phlorotannins, antifungal, larvicidal, brown seaweeds, biological activities

## Abstract

Phlorotannins are secondary metabolites produced by brown seaweeds with antiviral, antibacterial, antifungal, and larvicidal activities. Phlorotannins’ structures are formed by dibenzodioxin, ether and phenyl, ether, or phenyl linkages. The polymerization of phlorotannins is used to classify and characterize. The structural diversity of phlorotannins grows as polymerization increases. They have been characterized extensively with respect to chemical properties and functionality. However, review papers of the biological activities of phlorotannins have focused on their antibacterial and antiviral effects, and reviews of their broad antifungal and larvicidal effects are lacking. Accordingly, evidence for the effectiveness of phlorotannins as antifungal and larvicidal agents is discussed in this review. Online databases (ScienceDirect, PubMed, MEDLINE, and Web of Science) were used to identify relevant articles. In total, 11 articles were retrieved after duplicates were removed and exclusion criteria were applied. Phlorotannins from brown seaweeds show antifungal activity against dermal and plant fungi, and larvicidal activity against mosquitos and marine invertebrate larvae. However, further studies of the biological activity of phlorotannins against fungal and parasitic infections in aquaculture fish, livestock, and companion animals are needed for systematic analyses of their effectiveness. The research described in this review emphasizes the potential applications of phlorotannins as pharmaceutical, functional food, pesticide, and antifouling agents.

## 1. Introduction

Seaweeds are abundant in coastal regions and have become valuable sources of biologically active compounds and secondary metabolites, such as agar, carrageenan, alginate, alkaloids, phenolics, and phlorotannins, with extensive practical applications [1]. Phlorotannins are highly hydrophilic compounds formed by the acetate–malonate pathway. They contain phloroglucinol (Figure 1) (1,3,5-tryhydroxybenzene) units and have molecular sizes of 126 Da–650 kDa [2].

*Ishige okamurae*, *Ecklonia cava*, *E. kurome*, *E. stolonifera*, *Pelvetia siliquosa*, *Eisenia arborea*, and *E. bicyclis* as well as species in the genera *Cystophora* and *Fucus* have been reported to contain phlorotannins. Purified phlorotannins from these brown algae have antioxidant, antitumor, anticancer, anti-inflammatory, antiviral, antimicrobial, antifungal, and larvicidal activities, which are beneficial properties for the development of new functional agents [3,4,5,6,7].

Increasing antibiotic resistance and the spread of new variants of viruses are growing global problems [8]. Additionally, increases in mosquito larvae causing malaria, dengue hemorrhagic fever, filariasis, and chikungunya as well as biofouling marine invertebrate larvae have become major issues. Accordingly, the search for novel natural compounds to resolve these issues has been a major focus of research. Bioactive phlorotannins derived from brown algae have promising pharmacological and inhibitory effects [5,9,10,11,12,13] and, as described previously [10,14,15,16], may be valuable compounds for resolving these growing issues.

The five review papers on biological activities of phlorotannins reported by Eom et al. [5] focused on the antimicrobial activity of phlorotannins. Besednova et al. [17] and Zaporozhets and Besednova [18] have reviewed antiviral activities of phlorotannins. Nonetheless, reviews of their other biological activities, such as antifungal and larvicidal activities, are lacking. Accordingly, this review provides a comprehensive overview of antifungal and larvicidal activities of phlorotannins, providing a strong basis for their development as new functional agents. The biological activities of phlorotannins further support the utility of brown seaweeds as sources of novel functional agents derived from natural compounds.

## 2. Phlorotannins

Phlorotannins are produced and found in physodes, which are located in cells’ periphery and perinuclear regions [19]. Phlorotannins belong to phloroglucinol’s oligomers that can act as both primary and secondary metabolites. They are only found in brown seaweed and formed by the acetate–malonate (polyketide) pathway in the Golgi apparatus [20]. A combination of ether and phenyl, ether, dibenzodioxin, or phenyl linkages form the structures of phlorotannins (Figure 2). As a result, based on the structural linkage, phlorotannins can be divided into six groups. Eckols contain dibenzo-1,4-dioxin linkages, carmalols contain dibenzodioxin moiety, fucols contain aryl–aryl bonds, phloretols contain aryl–ether bonds, fucophloretols contain ether or phenyl linage, and fuhalols contain ortho-/para-arranged ether bridges containing an additional hydroxyl group on one unit [21].

Phlorotannins have been isolated from brown seaweed such as *Ecklonia cava*, *E. stolonifera*, *Sargassum ringgoldianum*, *Ishige okamurae*, *Fucus vesiculosus*, and *Eisenia bicyclis*, as well as species in the genera *Cystophora* and *Fucus*. Eckol, phloroglucinol, dieckol, diphlorethohydroxycarmalol, 6,6′-bieckol, phlorofucofuroeckol A, dioxinodehydroeckol, and 7-phloroeckol have been extracted from these seaweeds. Table 1 summarize the phlorotannins that were extracted from brown seaweeds.

## 3. Antifungal Activity of Phlorotannins

The antifungal activity of phlorotannins has been evaluated by Kim et al. [29], Lopes et al. [30], Lee et al. [31], and Corato et al. [32]. These studies have reported the antifungal activity of phlorotannins against dermal fungi, such as *Candida albicans*, *Epidermophyton floccosum*, *Trichophyton rubrum*, and *Trichophyton mentagrophytes*, and plant fungi, such as *Botrytis cinerea* and *Monilinia laxa*, as summarized in Table 2.

The effects of phlorotannins against dermal fungal pathogens have been evaluated. Lopes et al. [30] extracted phlorotannins from *Cystoseira nodicaulis*, *Crassiphycus usneoides*, and *Fucus spiralis* using *n*-hexane and then extracted using acetone:water (7:3). These phlorotannins exhibit antifungal activity against *C. albicans*, *E. floccosum*, and *T. mentagrophytes*. The MIC values of phlorotannins against these fungi range from 3.9 to 31.3 mg/mL. Fucofuroeckol-A, isolated from *Eisenia bicyclis*, and dieckol, isolated from *Ecklonia cava*, have antifungal activities [29,31]. Fucofuroeckol-A shows an MIC of 512 μg/mL against *C. albicans* [29], whereas dieckol exhibits a MIC of 200 μM against *Trichophyton rubrum* [31]. Although dieckol has shown antifungal activity, the concentration was extremely high. A general lack of selectivity of new drugs candidate should have >50% inhibition at a concentration less than 30 μM [33]. Corato et al. [32] have shown that phlorethols and fucophloretols extracted from *Laminaria digitata* are effective against plant fungal pathogens, such as *B. cinerea* and *M. laxa*, with 100% mycelial growth inhibition.

In fungal cell, phlorotannins block dimorphic complexes, resulting in the appearance of pseudohyphae with decreasing surface adhesive properties. The virulence and capacity to invade fungal host cells are also decreased by phlorotannins. On the other hand, phlorotannins induced reactive oxygen species (ROS) production and triggered early apoptosis, resulting in the activation of the CaMCA1 gene (Metacaspase 1) and membrane disruption. These inhibitory effects promote phlorotannins as new antifungal agents [29,30,32].

The effectiveness of phlorotannins as antifungal agents depends on numerous factors, such as the complex interactions between chemical compounds and the host and rates of degradation, hydrolysis, and polymerization. In the first step of nature compound discovery as new drug candidate, MICs are usually the starting point for larger preclinical evaluations of novel drug agents, and to ensure that compounds efficiently increase the success of treatment [31,32].

Increased rates of fungal infections in humans, animals, and plants necessitate the development of new antifungal agents. The antifungal effects of phlorotannin extracts, phlorethols, fucophloretols, fucofuroeckol-A, and dieckol have been evaluated. However, other subclasses of phlorotannins remain to be explored and should be a focus of further research aimed at the identification of novel antifungals.

## 4. Larvicidal Activity of Phlorotannins

The larvicidal activity of phlorotannins has been reported by Thangam and Kathiresan [34], Ravikumar et al. [35], Manilal et al. [36], Birrell et al. [37], Brock et al. [38], Lau and Qian [39], and Tsukamoto et al. [40]. These studies evaluated effects against mosquito larvae, such as *Aedes aegypti* and *Culex quinquefasciatus*, and against marine invertebrate larvae, such as *Acropora millepora*, *Balanus improvises*, *Hydroides elegans*, *Halocynthia roretzi*, and *Ciona savignyi*, as summarized in Table 3.

Phlorotannins show potential activity against mosquito larvae. Thangam and Kathiresan [34], Ravikumar et al. [35], and Manilal et al. [36] have reported that phlorotannins extracted from brown seaweeds, such as *Dictyota dichotoma*, *Lobophora variegata*, *Stoechospermum marginatum*, and *Sargassum wightii*, exhibit LC_50_ values ranging from 0.0683 to 85.11 μg/mL against mosquito larvae—namely, *A. aegypti* and *C. quinquefasciatus*. 

Birrell et al. [37] reported that phlorotannins from *Padina* sp. reduce the settlement of *Acropora millepora* larvae by 30%. Furthermore, phlorotannins from *Fucus vesiculosus* inhibit the larval settlement of *Balanus improvises* [38]. 

Study by Lau and Qian [39] reported that phlorotannins extract from *Sargassum tenerrimum* showed larvicidal activity in *Hydroides elegans* with an LC_50_ of 13.98 μg/mL. Tsukamoto et al. [40] demonstrated that phlorotannins extract inhibit 33% and 27% larval metamorphosis of *Ciona savignyi* and *Halocynthia roretzi* at low concentrations (25 μg/mL). 

In mosquito larvae, acute mortality and sublethality are the two main effects observed. With respect to sublethal effects, morphogenetic and external structural changes occur during the exposure period [41]. Other toxic effects, such as effects on growth, development, fecundity, fertility, and adult longevity in mosquitoes, have also been recorded [42,43]. Moreover, inhibitory effects on the cholinesterase enzymes cholinergic and gamma-aminobutyric acid (GABA) as well as mitochondrial and octopaminergic systems have also been recorded [44,45].

As larvae settlement inhibition agents, phlorotannins can influence the coral larval settlement process. In nature, phlorotannins delay the settlement process before larvae attach to substrates, even in areas free of macroalgae or with suitable substrates [37]. Furthermore, phlorotannins can inhibit settlement process of cyprids larvae. These findings indicate that phlorotannins from brown seaweeds might serve an essential ecological role as inhibitors of fouling. The larvicidal effects of phlorotannins might be mediated by various mechanisms, including the direct inhibition of the settlement and/or survival of larvae and regulation of the growth of bacterial microfoulers, affecting larval settlement. On the other hand, phlorotannins can quicken the metamorphosis of *Ciona savignyi* and *Halocynthia roretzi* compared to sulfoquinovosyl diacylglycerol at the same concentrations [40]. These findings suggest that phlorotannins can act as an antifouling agent without causing disruption to other organisms. 

However, most studies of the larvicidal activity of phlorotannins have focused on crude phlorotannins. To the best of our knowledge, other subclasses of phlorotannins, such as fuhalols, phlorethols, fucols, and fucophloroethols, have not been tested. These phlorotannins have a wide range of biological activities and further studies should evaluate their larvicidal effects and underlying mechanisms.

## 5. Extraction of Phlorotannins from Brown Seaweeds

Solid–liquid extraction using organic solvents is the most common method for obtaining phlorotannins from brown seaweeds. Phlorotannins can be extracted using polar solvents, including acetone, ethanol, and methanol. A mixture of polar solvents and water is often used to extract phlorotannins [46,47,48,49,50,51]. During the extraction procedure, the temperature is set to no more than 52 °C (and commonly to room temperature) to minimize the degradation of polyphenolic compounds [46,47]. The amount of phlorotannins extracted depends on the type of seaweed and the solvent used. Table 4 show phlorotannin yields obtained using organic solvents. Extraction of phlorotannins using both methanol:water (60%:40%) and methanol yielded phlorotannins ranging from 2 to 370 mg/g. Methanol solvent yielded the most phlorotannins but needs further processing to purify the compounds.

Naturally, concentration of phlorotannins in brown seaweeds is affected by biological factors, such as the species, tissue type, size, and age, as well as environmental conditions, such as nutrient levels, water temperature, season, herbivore intensity, and light intensity [46,47]. The extraction method also affects the yield.

Although solid–liquid extraction has been used to obtain phlorotannins from brown seaweeds, this method has a number of weaknesses, such as long extraction times for high yields, a lack of specificity, and the need to purify the extract [46,47,48]. Supercritical fluid extraction, microwave extraction, liquid extraction under pressure, ultrasonic extraction, and enzymatic extraction are alternative methods for phlorotannins extraction. These methods can increase yield, increase purity, and reduce extraction times [47,48,49,51,56].

Enzymatic extraction offers high yield values by the destruction of the cell wall. Puspita et al. [51] obtained a higher phlorotannins yield from *Sargassum polycystum* by the enzymatic method (21–38% phlorotannins) than by the solid solid–liquid method (3–15%). Similar to the enzymatic extraction method, the ultrasonic extraction method enables a high yield by destroying cell walls using mass transfer during the process [48]. Furthermore, the low time requirement is the greatest advantage of high-pressure liquid extraction and microwave methods [46,47,48,49,51].

## 6. Future Prospects for Phlorotannins

Since phlorotannins possess many biological activities, these compounds have attracted substantial research attention. The high effectiveness and low toxicity of these compounds support their utilization as components of pharmaceuticals, cosmetics, and food products (Figure 3).

According to Paradis et al. [57], Baldrick et al. [58], and Shin et al. [59], no side effects of phlorotannins have been recorded after testing in humans. Negara et al. [60] further reported that phlorotannins exhibit biological activities with low toxic effects on humans and animals. Phlorotannins successfully decrease the incremental areas under the curve in plasma insulin, cholesterol (both low-density and high-density lipoprotein levels), DNA damage, body fat ratio, and waist/hip ratio. Um et al. [61] reported no serious side effects, such as nausea, mild fatigue, abdominal distension, and dizziness. Thus, phlorotannins are new candidates for applications as pharmaceutical, food, pesticide, antibiofouling, and repellent agents.

Kim et al. [29], Lopes et al. [30], and Lee et al. [31] have shown that phlorotannins exhibit antifungal activities against dermatophytic fungi, such as *Candida albicans*, *Epidermophyton floccosum*, *Trichophyton rubrum*, and *Trichophyton mentagrophytes*, which cause skin infections. Accordingly, phlorotannins are promising compounds for the development of dermal creams with antifungal effects. In addition, Corato et al. [32] reported that phlorotannins successfully inhibit the mycelia of plant fungal pathogens, suggesting that they are potentially new natural pesticides. In food, antifungal activities exhibited by phlorotannins could be developed as food preservatives.

The larvicidal activity of phlorotannins in mosquitos reported by Thangam and Kathiresan [34], Ravikumar et al. [35], and Manilal et al. [36] suggests that they may be effective mosquito repellent agents. Phlorotannins have shown effects against marine invertebrate larvae [37,38,39,40], suggesting that they are natural antifouling agents. Unlike heavy metals, which act as broad-spectrum toxins to both targeted and nontargeted marine organisms [62], the natural antifouling effects of phlorotannins showed specificity to the target organism.

Therefore, recent research clearly supports the use of phlorotannins as pharmaceutical, cosmetic, antifouling, and food preservation agents. However, in-depth studies of phlorotannins are needed to determine their precise effects.

## 7. Method

Following Systematic Reviews and Meta-Analyses (PRISMA) guidelines, various online databases (Web of Science, ScienceDirect, MEDLINE, and PubMed) were used for literature searches [63]. “Phlorotannins OR antifungal OR larvicidal OR activity OR biological OR in vitro” was used as the search strategy. English language and effectiveness were applied as filters. In total, 85 articles were collected. After filtering, 11 articles were reviewed.

## 8. Conclusions

Our review revealed that phlorotannins from brown seaweeds exhibit activities against dermal and plant fungi, and mosquito and marine organism larvae. These findings provide a basis for the development of phlorotannins as new functional foods, feeds, pharmaceuticals, and larvicidal agents. To the best of our knowledge, their effects against viral, microbial, and parasitic infections have not been evaluated in fish, livestock, and companion animals; further studies on the biological activities of phlorotannins in these organisms are needed.

## Figures and Tables

**Figure 1 marinedrugs-19-00223-f001:**
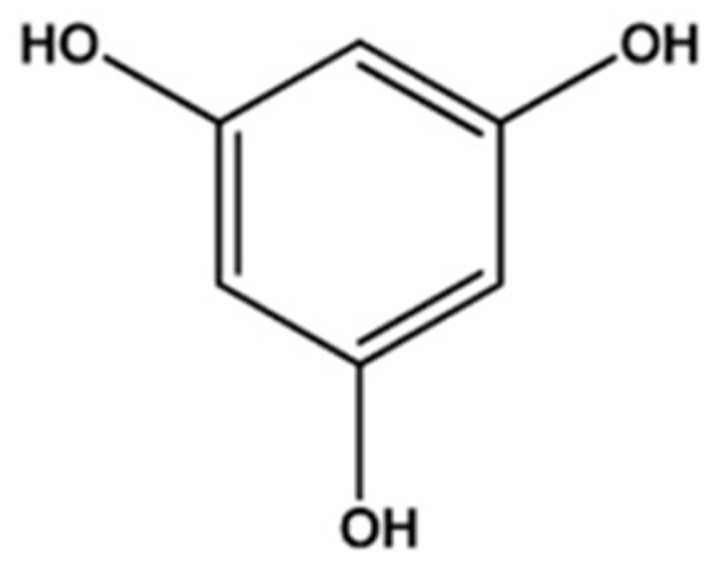
The basic structure of phlorotannins isolated from brown seaweeds [3].

**Figure 2 marinedrugs-19-00223-f002:**
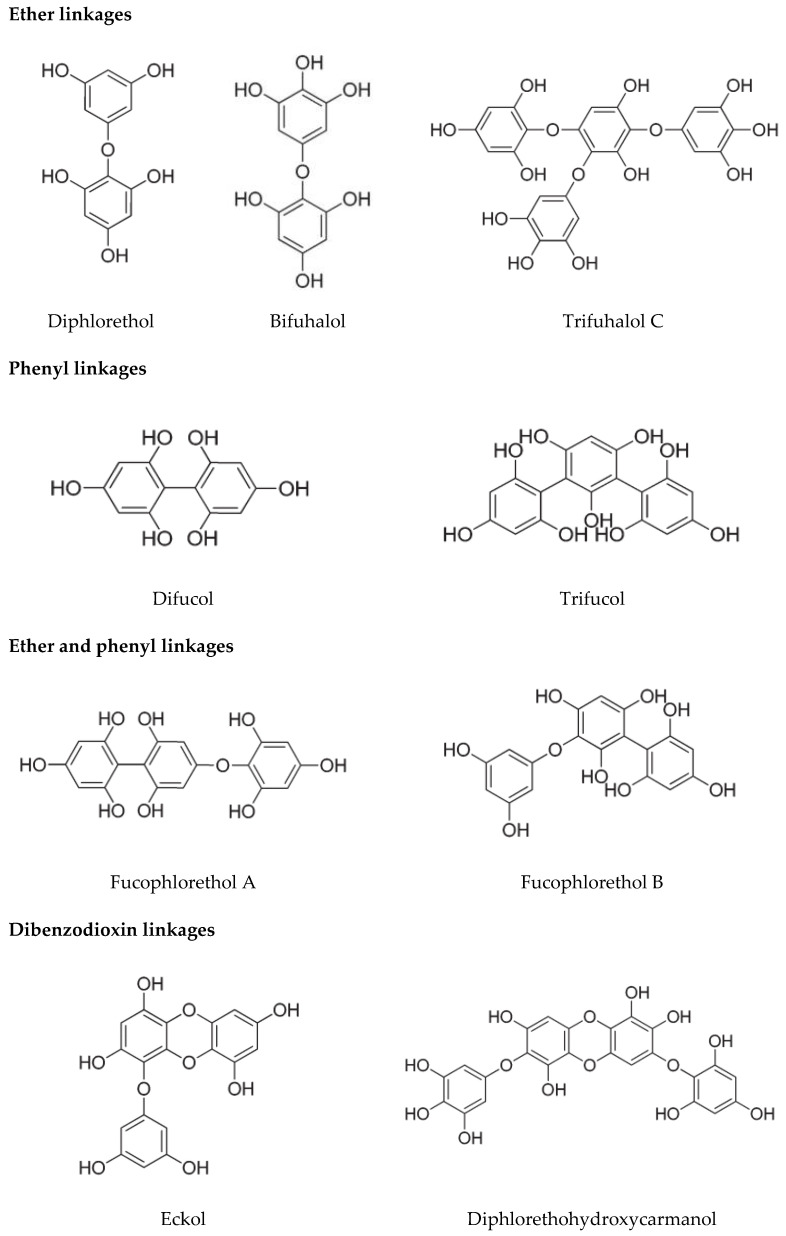
Structure of phlorotannins contain ether and phenyl-, ether-, dibenzodioxin-, or phenyl-linkages [22].

**Figure 3 marinedrugs-19-00223-f003:**
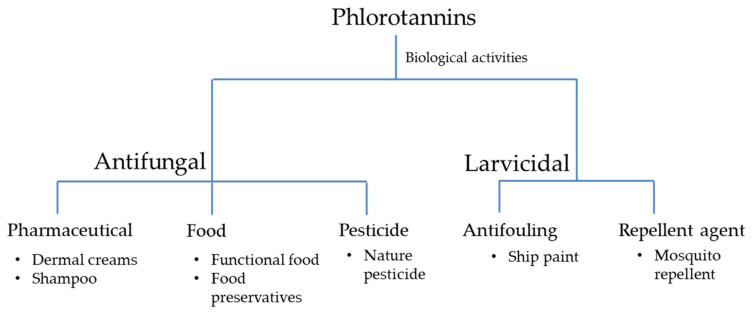
Application of phlorotannins as pharmaceutical, food, pesticide, antifouling, and repellent agents.

**Table 1 marinedrugs-19-00223-t001:** Phlorothannin compounds extracted from brown seaweeds.

Brown Seaweed	Compound	Ref.
*Ecklonia cava*	Eckol	[19,20,21]
	Phloroglucinol	[20,21,22,23]
	Dieckol	[20,21]
*Ecklonia stolonifera*	Phlorofucofuroeckol A	[24]
	Dieckol	
	Dioxinodehydroeckol	
*Eisenia bicyclis*	Phloroglucinol	[25]
	Eckol	
	7-phloroeckol	
	Phlorofucofuroeckol A	
	Dioxinodehydroeckol	
*Sargassum ringgoldianum*	Phlorotannins extract	[26]
*Ishige okamurae*	Phloroglucinol	[27]
	Diphlorethohydroxycarmalol	
	6,6′-bieckol	
*Fucus vesiculosus*	Phlorotannins extract	[28]

**Table 2 marinedrugs-19-00223-t002:** Antifungal activities of phlorotannins extracted from brown seaweeds.

Fungi	Extract/Chemical	Source	Activities	Ref.
**Dermal fungi**				
*Candida albicans*	Fucofuroeckol-A	*Eisenia bicyclis*	MIC ^a^ of 512 μg/mL	[29]
*Candida albicans*	Phlorotannins extract	*Cystoseira nodicaulis*	MIC of 15.6 mg/mL	[30]
*Candida albicans*		*Crassiphycus usneoides*	MIC of 31.3 mg/mL	
*Candida albicans*		*Fucus spiralis*	MIC of 31.3 mg/mL	
*Epidermophyton floccosum*		*Cystoseira nodicaulis*	MIC of 3.9 mg/mL	
*Epidermophyton floccosum*		*Crassiphycus usneoides*	MIC of 15.6 mg/mL	
*Epidermophyton floccosum*		*Fucus spiralis*	MIC of 7.8 mg/mL	
*Trichophyton rubrum*		*Cystoseira nodicaulis*	MIC of 3.9 mg/mL	
*Trichophyton rubrum*		*Crassiphycus usneoides*	MIC of 15.6 mg/mL	
*Trichophyton rubrum*		*Fucus spiralis*	MIC of 3.9 mg/mL	
*Trichophyton mentagrophytes*		*Cystoseira nodicaulis*	MIC of 7.8 mg/mL	
*Trichophyton mentagrophytes*		*Crassiphycus usneoides*	MIC of 31.3 mg/mL	
*Trichophyton mentagrophytes*		*Fucus spiralis*	MIC of 15.6 mg/mL	
*Trichophyton rubrum*	Dieckol	*Ecklonia cava*	MIC of 200 μM	[31]
**Plant fungi**				
*Botrytis cinerea*	Phlorethols	*Laminaria digitata*	MGI ^b^ of 100%	[32]
	Fucophloretols			
*Monilinia laxa*	Phlorethols			
	Fucophloretols			

^a^ MIC: Minimum inhibitory concentration. ^b^ MGI: Mycelia growth inhibition.

**Table 3 marinedrugs-19-00223-t003:** Larvicidal activities of phlorotannins extracted from brown seaweeds.

Larvae	Extract/Chemical	Sources	Activities	Ref.
**Mosquitos**				
*Aedes aegypti*	Phlorotannins extract	*Dictyota dichotoma*	LC_50_ ^a^ of 61.66 mg/L	[34]
*Aedes aegypti*	Phlorotannins extract	*Dictyota dichotoma*	LC_50_ of 0.0683 μg/mL	[35]
*Aedes aegypti*	Phlorotannins extract	*Lobophora variegata*	LC_50_ of 70.38 μg/mL	[36]
*Aedes* *aegypti*		*Stoechospermum marginatum*	LC_50_ of 82.95 μg/mL	
*Aedes* *aegypti*		*Sargassum wightii*	LC_50_ of 84.82 μg/mL	
*Culex quinquefasciatus*		*Lobophora variegata*	LC_50_ of 79.43 μg/mL	
*Culex quinquefasciatus*		*Stoechospermum marginatum*	LC_50_ of 85.11 μg/mL	
*Culex quinquefasciatus*		*Sargassum wightii*	LC_50_ of 87.09 μg/mL	
**Marine invertebrate**				
*Acropora millepora*	Phlorotannins extract	*Padina* sp.	30% of coral settlement was reduced	[37]
*Balanus improvisus*	Phlorotannins extract	*Fucus vesiculosus*	Larvae settlement was deterred at 31.5 μg/mL of concentration	[38]
*Hydroides elegans*	Phlorotannins extract	*Sargassum tenerrimum*	LC_50_ of 13.98 μg/mL	[39]
*Ciona savignyi*	Phlorotannins extract	*Sargassum thunbergii*	33% of larval metamorphosis were inhibited at 25 μg/mL	[40]
*Halocynthia roretzi*			27% of larval metamorphosis were inhibited at 25 μg/mL	

^a^ LC_50_: Lethal concentration.

**Table 4 marinedrugs-19-00223-t004:** Yield of phlorotannins extracted from brown seaweeds using organic solvent.

Sources	Solvent	Yield	Ref.
*Ascophyllum nodosum*	Methanol:Water (60%:40%)	2 mg/g	[52]
*Fucus serratus*		2.6 mg/g	
*Fucus vesiculosus*		2.92 mg/g	
*Laminaria hyperborean*		2.46 mg/g	
*Pelvetia canaliculata*		2.2 mg/g	
*Ascophyllum nodosum*	Methanol:Water (60%:40%)	6.66 mg/g	[53]
*Himanthalia elongata*		2.79 mg/g	
*Ecklonia kurome*	Methanol	370 mg/g	[54]
*Ishige okamurae*	Methanol	190 mg/g	[55]

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
