# Peer review of "Antifungal and Larvicidal Activities of Phlorotannins from Brown Seaweeds"

_marinedrugs, 2021, doi:10.3390/md19040223_

Round 1

Reviewer 1 Report

This review by Negara et al focuses on the antimicrobial activities of phlorotannins. While this is an important class of natural products, I have the following major concerns:

  1. Although the authors claim that there are no reviews of their efficacy against organisms other than bacteria, a quick pubmed search found multiple published in the last two years. For example, PMC7755586 covers multiple phlorotannins with efficacy against coronaviruses and both PMC7460554 and PMC7921925 includes considerable information on their activity against multiple viruses. I would recommend that the authors focus on antifungal and larvicidal activities instead, which lack detailed reviews.
  2. Antivirals and antifungals are considered subclasses of antimicrobials. Authors should either only refer to antimicrobials or only to antivirals and antifungals in the title, abstract, etc. Antimicrobial should not be used as a synonym for antibacterial.
  3. On a related note, line 22, “antimicrobial activity against foodborne and dermal pathogens” is redundant with prior and subsequent lines on viruses and fungi, both of which are also pathogens. This issue also applies to lines 101-102 and Table 2. If the authors mean “bacteria” instead of “pathogens”, please use “bacteria” instead. Likewise, in Table 3, please change “pathogens” to “fungi”.
  4. Abstract usefulness could be improved if the authors described the structural characteristics of phlorotannins.
  5. Lines 35-36: “They contain phloroglucinol (1,3,5-tryhydroxybenzene) units and have molecular sizes of 126–650 kDa [2].”. Phrasing is confusing and should be changed to 126 Da-650 kDa, to make it clear that some phlorotannins are smaller (g tetrafucol A at 498 Da).
  6. Authors should include a figure showing representative structures.
  7. Lines 67-68: “However, at 50 μM, 8,4-dieckol from E. cava exhibits antiviral activity against HIV-1 with a 91% inhibition ratio”. Authors should be more critical when reporting such findings. From a drug development perspective, 50 μM is an extremely high dose and unlikely to be translatable to any clinical application. Such caveats should be mentioned. Same issue at lines 160-162.
  8. There are multiple instances of species and genera not being correctly italicized, throughout the manuscript (g. E. cava lines 64, 67, 69). Authors should correct this.
  9. Table 1, Table 2, Table 3, Table 4: “crude extracts” could contain other compounds beyond phlorotannins, which could be responsible for the biological activity. They should not be included in this manuscript. This is particularly true given the data in Table 5, which indicates a reliance on simple methanol or methanol:water extraction solvents, which will extract many other compounds.
  10. Have any of these compounds been tested in humans or animal models? Or have all the antimicrobial activities been assayed only in vitro? If the latter, such a limitation should be mentioned.

Minor issues:

  1. Table 1 typo: “Soruces" should be “sources”, “aimals” should be “animals”
  2. Table 1: “Viruses infected human”, “Viruses infected aimals”. I am assuming these compounds were not tested directly on humans or animals, and instead the authors mean “viruses infective to humans”, “viruses infective to animals” (or “human-infective viruses”, “animal-infective viruses”). Please correct.
  3. Table 1, Table 2, Table 3, Table 4: Correct “2.7 μM of IC50” to be “IC50 of 2.7 μM”, and similar issues throughout the tables.
  4. Lines 90-95: “To the best of our knowledge, the antiviral activity of phlorotannins has only been 90 tested against human viruses, such as HIV-1, measles virus, H1N1, and SARS-CoV 3CLpro, 91 as well as some animal viruses, such as murine norovirus and porcine epidemic diarrhea 92 virus. The antiviral activity of phlorotannins in agriculture, aquaculture, livestock 93 (chicken, duck, swine, and cattle), and companion animals (birds, cats, and dogs) has not 94 been reported.” This is a confusing statement as influenza viruses, for example, can infect multiple species and not just humans.
  5. Table 2: “Did not growth at”. Do the authors mean that the listed compounds caused 100% bacterial dose inhibition at the listed dose? If yes, please rephrase.
  6. Table 5 typo: “Yield” should be “Yield”.

Author Response

RESPONSE TO THE COMMENTS OF REVIEWER#1

This review by Negara et al focuses on the antimicrobial activities of phlorotannins. While this is an important class of natural products, I have the following major concerns:

We thank the Reviewer for the appreciation of our manuscript. We have carefully addressed all the concerns raised by the Reviewer, and made the necessary changes following your suggestions. Please note that the modifications made in the manuscript are written in red font.

Point 1: Although the authors claim that there are no reviews of their efficacy against organisms other than bacteria, a quick pubmed search found multiple published in the last two years. For example, PMC7755586 covers multiple phlorotannins with efficacy against coronaviruses and both PMC7460554 and PMC7921925 includes considerable information on their activity against multiple viruses. I would recommend that the authors focus on antifungal and larvicidal activities instead, which lack detailed reviews.

Point 2: Antivirals and antifungals are considered subclasses of antimicrobials. Authors should either only refer to antimicrobials or only to antivirals and antifungals in the title, abstract, etc. Antimicrobial should not be used as a synonym for antibacterial.

< Minor issues>

Point 1: Table 1 typo: “Soruces" should be “sources”, “aimals” should be “animals”

Point 2: Table 1: “Viruses infected human”, “Viruses infected aimals”. I am assuming these compounds were not tested directly on humans or animals, and instead the authors mean “viruses infective to humans”, “viruses infective to animals” (or “human-infective viruses”, “animal-infective viruses”). Please correct.

Point 4: Lines 90-95: “To the best of our knowledge, the antiviral activity of phlorotannins has only been 90 tested against human viruses, such as HIV-1, measles virus, H1N1, and SARS-CoV 3CLpro, 91 as well as some animal viruses, such as murine norovirus and porcine epidemic diarrhea 92 virus. The antiviral activity of phlorotannins in agriculture, aquaculture, livestock 93 (chicken, duck, swine, and cattle), and companion animals (birds, cats, and dogs) has not 94 been reported.” This is a confusing statement as influenza viruses, for example, can infect multiple species and not just humans.

Response: We thank the Reviewer for the insightful inputs. As the concerns raised in Points 1, 2, and point 1, 2, 4, and 5 of the “minor issues” section are related, we have decided to combine the response to these questions when addressing them in the manuscript. Considering your recommendations, we have deleted the sections on antiviral and antimicrobial activities.

Point 3: On a related note, line 22, “antimicrobial activity against foodborne and dermal pathogens” is redundant with prior and subsequent lines on viruses and fungi, both of which are also pathogens. This issue also applies to lines 101-102 and Table 2. If the authors mean “bacteria” instead of “pathogens”, please use “bacteria” instead. Likewise, in Table 3, please change “pathogens” to “fungi”.

Response 3: We thank the Reviewer for highlighting this points. We have replaced the word “pathogen” for a more specific word like “fungi” in all relevant places of the revised manuscript.

Point 4: Abstract usefulness could be improved if the authors described the structural characteristics of phlorotannins.

Response 4: We thank the Reviewer for the constructive suggestion. Accordingly, we have revised the abstract as follows:

“Phlorotannins are secondary metabolites with antiviral, antibacterial, antifungal, and larvicidal activities, produced by brown seaweeds. The structures of these metabolites are formed by dibenzodioxin, ether, and phenyl ether or phenyl linkages. Moreover, the polymerization of phlorotannins is used for classification and characterization purposes, and it is known that their structural diversity grows as polymerization increases. While their chemical properties and functionality have been extensively characterized, the review papers on the biological activities of phlorotannins have mainly focused on their antibacterial and antiviral effects, and not in their broad antifungal and larvicidal effects. Therefore, in this review evidence on the effectiveness of phlorotannins as antifungal and larvicidal agents is discussed. Online databases (ScienceDirect, PubMed, MEDLINE, and Web of Science) were used to identify relevant articles on the topic, and a total of 11 articles were retrieved after the duplicates were removed and the exclusion criteria were applied. Phlorotannins from brown seaweeds show antifungal activity against dermal and plant fungi, and larvicidal against mosquitos and marine invertebrates larvae. However, further studies of the biological activity of phlorotannins against fungal and parasitic infections in aquaculture fish, livestock, and companion animals are needed for systematic analyses of their effectiveness. The researches described in this review emphasize the potential applications of phlorotannins as pharmaceutical, pesticide, and biofouling agents, as well as functional foods.

Keywords: phlorotannins; antifungal; larvicidal; brown seaweeds; biological activities”

Point 5: Lines 35-36: “They contain phloroglucinol (1,3,5-tryhydroxybenzene) units and have molecular sizes of 126–650 kDa [2].”. Phrasing is confusing and should be changed to 126 Da-650 kDa, to make it clear that some phlorotannins are smaller (g tetrafucol A at 498 Da).

Response 5: We apologies for the confusion. To aid clarity, we have revised the text as follows:

“They contain phloroglucinol units (1,3,5-tryhydroxybenzene) and have a molecular sizes of 126 Da–650 kDa [2].”

Point 6: Authors should include a figure showing representative structures.

Response 6: In response to your comment, we have added the following figures: “Figure 1. The basic structure of phlorotannins isolated from brown seaweeds” and “Figure 2. Structure of phlorotannins containing ether and phenyl-, ether-, dibenzodioxin-, or phenyl- linkages”.

Point 7: Lines 67-68: “However, at 50 μM, 8,4-dieckol from E. cava exhibits antiviral activity against HIV-1 with a 91% inhibition ratio”. Authors should be more critical when reporting such findings. From a drug development perspective, 50 μM is an extremely high dose and unlikely to be translatable to any clinical application. Such caveats should be mentioned. Same issue at lines 160-162.

Response 7: We thank the Reviewer for the insightful inputs. We have revised the text as follows:

“dieckol exhibits a MIC of 200 μM against Trichophyton rubrum [24]. Although dieckol has shown antifungal activity, the concentration required was extremely high. General lack of selectivity in new‑drug candidates can be identified as >50% inhibition at a concentration lower than 30 μM [25].”

Point 8: There are multiple instances of species and genera not being correctly italicized, throughout the manuscript (g. E. cava lines 64, 67, 69). Authors should correct this.

Response 8: We have checked all the scientific names of organisms in the revised manuscript and corrected them

Point 9: Table 1, Table 2, Table 3, Table 4: “crude extracts” could contain other compounds beyond phlorotannins, which could be responsible for the biological activity. They should not be included in this manuscript. This is particularly true given the data in Table 5, which indicates a reliance on simple methanol or methanol:water extraction solvents, which will extract many other compounds.

Response 9: We apologies for the confusion and would like to clarify that the crude extracts we have mentioned in our manuscript are the crude phlorotannins.. To aid clarity, we have revised the text and replaced “crude extracts” for “crude phlorotannins” in all relevant places.

Point 10: Have any of these compounds been tested in humans or animal models? Or have all the antimicrobial activities been assayed only in vitro? If the latter, such a limitation should be mentioned.

Response 10: We have provided information in the suggested aspects by discussing future prospects for phlorotannins. The text added reads as follows:

“According to Paradis et al. [57], Baldrick et al. [58], and Shin et al. [59], no side effects of phlorotannins have been recorded after testing in humans. Negara et al. [60] further reported that phlorotannins exhibit biological activities with low toxic effects on humans and animals.  Phlorotannins successfully decrease the incremental areas under the curve in plasma insulin, cholesterol (both low-density and high-density lipoprotein levels), DNA damage, body fat ratio, and waist/hip ratio. Um et al. [61] reported no serious side effects, such as nausea, mild fatigue, abdominal distension, and dizziness. Thus, phlorotannins are new candidates agents for applications in the pharmaceutical industry.”

< Minor issues>

 Point 3: Table 1, Table 2, Table 3, Table 4: Correct “2.7 μM of IC50” to be “IC50 of 2.7 μM”, and similar issues throughout the tables.

Response 3: We thank the Reviewer for accurate observations. We have replaced “2.7 μM of IC50” for “IC50 of 2.7 μM at all the relevant instances.

Point 5: Table 2: “Did not growth at”. Do the authors mean that the listed compounds caused 100% bacterial dose inhibition at the listed dose? If yes, please rephrase.

Response 5 We thank the Reviewer for the helpful comments, we have revised the text as follows: “100% inhibition at”.

Point 6: Table 5 typo: “Yield” should be “Yield”.

Response 6: We have corrected the mentioned misspelling throughout the manuscript.

Reviewer 2 Report

The article „Antiviral, Antimicrobial, Antifungal, and Larvicidal Activities 2 of Phlorotannins from Brown Seaweeds” is a narrative review. The authors collected articles regarding the antimicrobial effects of phlorotannins. It seems that there are not many papers published on this field (this paper collected 26 of them); regardless, collecting and reviewing the available data is valuable to researchers worldwide. The investigation of antimicrobial effects of phlorotannins may prove to be important because of the prevalence of multiresistant bacteria and novel viruses. However, this review suffers from several critical flaws.

The data collected in the tables have great value to the readers. However, the text in the article only repeats what we can already read out from the tables. Therefore, the text does not answer specific questions about the topic and the authors do not provide detailed expert opinion either. The authors should use this data to answer or discuss specific questions on this topic (e.g. What is the mechanism of action of these compounds? Are they better than compounds already in use? Is it possible to implement them in therapy? What are the limitations?)

The paragraph explaining the extraction methods does not fit in to the other paragraphs.

The description of search methodology is missing from the article. The authors mention it in the abstract that they found 26 articles during their search after applying some sort of exclusion criteria, but they do not elaborate on that in the article.

The authors use antimicrobial and antibacterial as synonyms, which is not correct.

The definitions of EC50, IC50 and LC50 are oversimplified in the article.

Author Response

RESPONSE TO THE COMMENTS OF REVIEWER#2

 The article “Antiviral, Antimicrobial, Antifungal, and Larvicidal Activities 2 of Phlorotannins from Brown Seaweeds” is a narrative review. The authors collected articles regarding the antimicrobial effects of phlorotannins. It seems that there are not many papers published on this field (this paper collected 26 of them); regardless, collecting and reviewing the available data is valuable to researchers worldwide. The investigation of antimicrobial effects of phlorotannins may prove to be important because of the prevalence of multiresistant bacteria and novel viruses. However, this review suffers from several critical flaws.

Point 1: The data collected in the tables have great value to the readers. However, the text in the article only repeats what we can already read out from the tables. Therefore, the text does not answer specific questions about the topic and the authors do not provide detailed expert opinion either. The authors should use this data to answer or discuss specific questions on this topic (e.g. What is the mechanism of action of these compounds? Is it possible to implement them in therapy? What are the limitations?)

Response 1: We thank the Reviewer for the insightful suggestion. Following your comment, we have thoroughly revised the bibliography to discuss the questions that you have mentioned.

 Point 2: The paragraph explaining the extraction methods does not fit in to the other paragraphs.

Response 2: As per your comment, we have revised the extraction methods section.

Point 3: The description of search methodology is missing from the article. The authors mention it in the abstract that they found 26 articles during their search after applying some sort of exclusion criteria, but they do not elaborate on that in the article.

Response 3: We apologies for the reduce information on that topic. We have added information on the search methodology in the method section.

Point 4: The authors use antimicrobial and antibacterial as synonyms, which is not correct.

Response 4: This concern has also been expressed by Reviewer 1; Thus, we have deleted the sections on antiviral and antimicrobial activities.

Point 5: The definitions of EC50, IC50 and LC50 are oversimplified in the article.

Response 5: We thank the Reviewer for the helpful comment. We have revise the manuscript to improve EC50, IC50 and LC50 definitions.

Reviewer 3 Report

The Authors in the manuscript entitled: “Antiviral, Antimicrobial, Antifungal, and Larvicidal Activities of Phlorotannins from Brown Seaweeds” reported biological activities  of phlorotannins.

The manuscript is interesting; the authors must italicize all the names of algae and microorganisms

The manuscript is interesting; the authors must write in italicize all the names of algae and microorganisms. In addition, the names of the algae must be standardized; i.e. they must be all either abbreviated (e.g. E. cava) or all in full (e.g. Ecklonia cava) (page 1 lines 37 and 38), (page 2 line 71; page 3 lines 83, 85, 88, 105, 106). etc .......

The authors must also revised the name of algae reported in the tables.

Author Response

RESPONSE TO THE COMMENTS OF REVIEWER#3

 The manuscript is interesting; the authors must italicize all the names of algae and microorganisms.

We thank the Reviewer for the positive comment on our manuscript.. We have carefully addressed all the concerns raised by the Reviewer and made the necessary changes following the suggestions provided. The changes made in the manuscript are shown in red font.

Point 1: The manuscript is interesting; the authors must write in italicize all the names of algae and microorganisms. In addition, the names of the algae must be standardized; i.e. they must be all either abbreviated (e.g. E. cava) or all in full (e.g. Ecklonia cava) (page 1 lines 37 and 38), (page 2 line 71; page 3 lines 83, 85, 88, 105, 106). Etc. The authors must also revised the name of algae reported in the tables.

Response 1: We thank the Reviewer for the valuable comment. We have corrected and checked all the scientific names of organisms in the revised manuscript as suggested.

Round 2

Reviewer 1 Report

All my comments have been addressed satisfactorily.

Author Response

Dear, Reviewer 1.

Thank you for your valuable comments. The English in this article has been checked by a native English speaker. For a certificate, please see the attached file.

Best regards

Prof. Jae-Suk Choi

Reviewer 2 Report

After the revisions, the article improved significantly. I only have one minor comment:

"Accordingly, this review provides a comprehensive overview of the biological activities of phlorotannins, including not only antimicrobial but also antiviral, antifungal, and larvicidal activities, providing a strong basis for their development as new functional agents."
The antimicrobial and antiviral overview is not included in the article any more.

Author Response

Dear, respected Reviewer 2. 

Thank you for your comment. 

In response to  Reviewer 1' comments (please see below), we have inevitably deleted  section on antiviral and antimicrobial activities in the revised manuscript. 

I earnestly ask for your understanding.

Kind regards,

Prof. Jae-Suk Choi

....................................................................................................................................................................

Reviewer 1' comments.

1. Although the authors claim that there are no reviews of their efficacy against organisms other than bacteria, a quick pubmed search found multiple published in the last two years. For example, PMC7755586 covers multiple phlorotannins with efficacy against coronaviruses and both PMC7460554 and PMC7921925 includes considerable information on their activity against multiple viruses. I would recommend that the authors focus on antifungal and larvicidal activities instead, which lack detailed reviews.

2. Antivirals and antifungals are considered subclasses of antimicrobials. Authors should either only refer to antimicrobials or only to antivirals and antifungals in the title, abstract, etc. Antimicrobial should not be used as a synonym for antibacterial.